# Exploring Partners, Parenting and Pregnancy Thinking in Late Adolescents and Young Adults with Inherited Metabolic Disorders

**DOI:** 10.3390/pediatric17030056

**Published:** 2025-05-13

**Authors:** Albina Tummolo, Giulia Paterno, Rosa Carella, Livio Melpignano, Donatella De Giovanni

**Affiliations:** 1Department of Metabolic Diseases, Clinical Genetics and Diabetology, Giovanni XXIII Children Hospital, Azienda Ospedaliero-Universitaria Consorziale, 70126 Bari, Italy; giulia.paterno@policlinico.ba.it (G.P.); rosa.carella@policlinico.ba.it (R.C.); donatella.degiovanni@policlinico.ba.it (D.D.G.); 2Medical Direction, Giovanni XXIII Children Hospital, Azienda Ospedaliero-Universitaria Consorziale, 70126 Bari, Italy; livio.melpignano@policlinico.ba.it

**Keywords:** inherited metabolic disorders, partner choice, parenthood, fertility, pregnancy

## Abstract

Introduction: The psychosocial impact of living with an Inherited Metabolic Disorder (IMD) is becoming increasingly relevant and can have a significant impact on planning the future, conditioning the reproductive decisions made during adolescence and young adulthood. The aim of this paper is to explore thoughts about partner choices, parenthood and pregnancy among adolescents and young adults affected by IMDs. Methods: A cross-sectional study was performed. A sample of 23 patients affected by a range of IMDs were interviewed. Twenty-two questions were provided, distinguished into four main themes: partners, parenthood, pregnancy and need for information. Results: More than half of participants (57%) reported insecurities about relationships and declared that they were single for this reason, with most (70%) having a hope of having children in the future, although with the awareness and fear that they could also be affected. Almost all females (90%) consider themselves able to carry a pregnancy in a way similar to other women. There was the common need for more information about their potential fertility and parenthood linked to their condition. Conclusion: Being diagnosed with an IMD can influence personal decisions regarding relationships and reproduction. The early identification of issues in these domains may enhance referrals for personalized interventions and build more focused support programmes.

## 1. Introduction

Inherited metabolic disorders (IMDs) are a group of genetic conditions where the body’s metabolism is disrupted due to defects in specific enzymes or proteins that are responsible for converting food into energy or for processing specific molecules in the body. These disorders are usually inherited in an autosomal recessive pattern, meaning that both copies of the gene in question must have mutations for the disorder to manifest.

IMDs can affect various metabolic pathways, including those responsible for breaking down proteins, fats and carbohydrates, as well as synthesizing essential molecules, such as hormones or neurotransmitters [1].

IMDs were first described by Sir Garrod in 1901 [2] and nowadays are increasingly recognised due to improved diagnostic technologies [3,4,5] and worldwide newborn screening programmes [6].

The literature provides robust evidence of improved treatment options for many IMDs [7,8], most of which were once deemed untreatable. As survival and quality of life improve, attention has shifted to new challenges in managing psychosocial aspects of living with IMDs [9,10,11].

Beyond the obvious health challenges related to a genetic disorder, patients with IMDs also experience psychosocial difficulties and challenges, as their day-by-day life may be strongly impacted not only by the disease’s physical consequences, but also by the psychological stress of disease management and therapy [12,13,14]. This situation may have an effect on the social interactions and relationships of young individuals with their peers [15,16] and on the decision-making process regarding partner choices and reproductive decisions. Furthermore, many studies have provided evidence of possible genetic influences on several aspects of sexual behaviour, such as the desire to become parents and the propensities for marriage [17].

Thoughts about finding a partner and becoming parents become increasingly important, occupying a significant part of their future planning and expectations, from adolescence onwards [18,19]. In early adulthood, relationship building and family planning become increasingly relevant [20,21].

Although the psychological issues of young patients with genetic disorders are becoming increasingly recognized in clinical practice in recent years, evidence regarding partner perceptions, choices and reproductive decision making are still scarce and require more specific and detailed information. In a recent clinical survey on young adults affected by sickle cell disease, their reproductive experiences and intentions were analysed. Most (68%) were sexually active, and 45% reported contraception use; 92% did not desire a pregnancy in the near future [22].

In a qualitative study conducted by Severijns et al. on couples at risk of transmitting a genetic disease, feelings of guilt and fear were commonly reported and more in-depth decision support was required by patients [23].

Pregnancy-related complications and the risk of disease transmission may discourage reproductive decisions in young women with IMDs [24,25]. Phenylketonuria (PKU), one of the most common IMDs [26], represents a paradigm of this issue as it adds to the risk of transmissibility of the disease, along with the potential teratogenicity of Phenylalanine, during pregnancy, thus strongly influencing the woman’s decisions [27,28].

Despite the improvements in medical care for women with genetic disorders leading to an increased number of women reaching reproductive age (Harris) [25], female patients affected by IMDs have been reported to be at risk of acute metabolic decompensation during catabolic stress periods, including labour, delivery and the postpartum period [29].

We hypothesized that young adults affected by IMDs may face greater difficulties in forming relationships and planning a family due to disease management burdens and inheritance concerns.

This study aimed to explore perceptions about partners, parenthood and pregnancy, while also expanding on the need for information and the preferred way to receive information, among adolescents and young patients affected by different forms of IMDs.

## 2. Materials and Methods

### 2.1. Design, Setting and Sample

This was a cross-sectional study based on a telephone survey conducted from July 2024 to September 2024 at the Division of Inherited Metabolic Disorders, Clinical Genetics and Diabetology, of Children Hospital Giovanni XXIII in Bari, Italy. This study was part of the support programme for adolescents’ health and social wellbeing held by the hospital. As part of the hospital support programme, a genetic counselling was offered to those who required it, also including contraceptive counselling if necessary.

This study formed part of the hospital’s support programme for adolescent health and social well-being. Genetic and contraceptive counselling were offered as needed.

Inclusion criteria were as follows: males and females followed up for IMD, aged between 16 and 35 years. Low cognitive level, mild forms of the diseases, not requiring any treatment, irregular follow-up and diagnosis not confirmed by genetic analysis represented exclusion criteria. Participants were recruited by an invitation during a regular outpatient visit. Informed consent was obtained prior to telephone interviews.

### 2.2. Data Collection

Google Forms [30] was used to administer the survey, which included 22 questions across four themes: expectations about having a partner and/or a spouse; propensity to have children in the future; capability to face a pregnancy (for female participants only). The questions also addressed whether they would like to receive more information and counselling about this context. Some samples of questions and related themes are reported in Table 1.

Both open- and close-ended questions were used. Participants could elaborate using optional text boxes. Telephone interviews were conducted by a single investigator. The interviews were based on a broad scoping review of the literature [31] and revised through discussion among investigators, all with clinical experience in following and treating IMDs. The survey was delivered for 2 months; the last interview was performed on 14 September 2024.

Data were anonymized using alphanumeric codes. Descriptive statistical analysis was performed using Microsoft^®^ Excel 2016 MSO, version 2016. Continuous variables were expressed as mean (±standard deviation, SD) or median and range, depending on their distribution; categorical variables were reported as frequencies and percentages.

This study was conducted according to the guidelines of the “Declaration of Helsinki” and received ethics approval from the Local Ethics Committee IRCCS Istituto Oncologico “Gabriella Serio”, Bari, (Protocol No.: 1643/CEL).

## 3. Results

Thirty-two patients were initially contacted. Five did not respond after initial contact, and one declined participation. Twenty-three eligible patients completed the interviews, resulting in a 71% response rate.

Individual open and close-ended questions were asked, each lasting on average 15 min (between 10 and 20 min). The age range of the study participants was 18–32; the majority were males (56%) and affected by PKU (56%). Table 2 reports their main demographic characteristics and type of disease. Genetic confirmation was available, and the results from the genetic analysis are reported as Appendix A.

All participants had an IQ above 70 (minimum IQ was 71) and no motor or intellectual disabilities were reported. Additionally, 19 out of 23 (82%) individuals completed high school, 5 (21%) were attending university at the time of the survey, 3 were employed and 4 were neither studying nor working. Twenty-one (91%) individuals lived with their families; the remaining three males had either moved in with their partners or relocated for university.


**Theme 1: Partners**


Most of the participants (20/23, 87%) expressed an interest in a couple’s relationship, but less than half of them (10/23, 43%) declared that they have a partner, of whom 5 were female (50% of all females) and 5 were male (38% of all males). Four participants (one female and three males) had never had a partner; their mean age was 20.7 ± 2 years. None of the participants were married at the time of the survey. The mean age of those in a relationship was 25.3 years, compared to 23.4 years for single participants. Thirteen participants (57%) declared that they were insecure about sentimental relationships and declared that they were single for this reason. Concerns about the fact that the disease could frighten the partner, and that their disease-linked lifestyle was not compatible with the life of a couple, represented the main reported limitations.

Most participants (13/23, 57%) said they would disclose their condition early in a new relationship, often at the first meeting, to build trust and ensure compatibility:
*“I would talk about my disease as soon as I have the opportunity to do it. I want to be sure that my boyfriend would be able to face the difficulties and challenges with me”.**(female, 27 years, urea cycle disorder)*


**Theme 2: Parenthood**


Most of the participants (16/23, 70%) hoped to have children in the future, and the remaining participants did not consider parenthood as a part of their future plans; the majority of this group were males (5), with a mean age of 24 ± 4 years. Concerns about reduced fertility were exclusively expressed by females (four), who felt that their condition might limit the possibility of conceiving:
*“I feel uncertain about the possibility of conceiving due to my disease, but I do not think I should be an inadequate parent”.**(female, 19 years PKU)*

Nearly half of all participants (12/23, 52%) expressed fear that their child might inherit the same disease, as they believe that it is likely that their genetic condition could be transmitted to sons. Therefore, they are aware that they face personal and ethical choices in the future:
*“If I decide to have a child, I would ask my partner to do the genetic test to discover if he is carrier of the disease”.**(female, 21 years, galactosemia)*


**Theme 3: Pregnancy**


Almost all of the females who participated (9/10, 90%), had optimistic expectations about being able to carry a pregnancy similarly to other women., Nonetheless, half of them (five) believed that special assistance would be required during their pregnancy due to potential increased risks, such as nausea, vomiting or more frequent monitoring:
*“I’m not sure I can cope with the restricted diet and phenylalanine monitoring during pregnancy. I think I would need help in this period”.**(24 years, PKU)*


**Theme 4: Information**


The vast majority (20/23, 87%) of participants expressed a desire to receive more information about fertility and parenthood related to their condition. To the same extent, they would like to share experience in this context with people affected by the same condition. Three (13%) were not interested in further information, as they did not foresee parenthood in their future plans. They were all males, aged 23, 30 and 32 years, and two of them had declared that they have a partner.
*“I live the relationship with my partner on a day-by-day basis and I’m afraid that facing this topic may alter the equilibrium between me and my girlfriend”.**(male, 22 years, maple syrup urine disease)*

The preferred sources for receiving information were websites (11), followed by dedicated meetings with the metabolic team (6) and finally meeting with other experts in the field (3) (gynaecologists, obstetrics and neonatologists). Of those preferring websites, eight (72%) participants were under 25 years of age, and eight were males.

Only one participant reported a negative view of online information:
*“I feel more confident in reading this information from a paper leaflet, because in this way I’m sure about the source and the authors”.**(male, 18 years, PKU)*

## 4. Discussion

This explorative cross-sectional study investigated, for the first time, thinking about partner choices, fertility and pregnancy among young subjects affected by IMDs. The results have pointed out some main findings: more than half of the participants felt insecure about sentimental relationships and declared that they were single for this reason, and most of the participants have a hope of having children in the future, although with the awareness and fear that they could be also affected. Fifty percent of the females had no partner at the time of this study; however, most of them consider themselves able to carry a pregnancy in a way similar to other women. There was a common need for more information about their potential fertility and parenthood linked to their condition, and the youngest patients preferred to receive this information by websites. A graphical representation of the key results of this study is reported in Figure 1.

**Alternative Text:** The figure provides a graphical representation of the key results of the study, with three pie charts and one bar chart. They represent, in different colors, the proportion of positive and negative responses about: having a partner, willing parenthood, being able to carry pregnancy, needing more information.

The perception and the choice of a potential partner for young individuals affected by IMDs can be complex and influenced by various factors, and it appears to represent a major limitation for our patients.

They may be hesitant to start a relationship with someone, as they may be concerned about the impact the disease management may have on the partner’s quality of life. In particular, many IMDs require patients to follow a restricted diet regimen [32,33] and patients may worry about how their potential partner and their social circle in general would perceive this limitation. The social impact of a strict diet regimen represents an issue for young subjects and is the main reason for diet discontinuation in PKU patients since adolescence [34,35]. The diet burden may play a role in the partnership of these young subjects. In this context, it has been reported that personal experiences with an IMD, including its physical and emotional impacts, may guide an individual toward the choice of a partner who has a deep understanding and empathy for their situation [36], as also referred to by some of our patients.

In our study, about one-third of the participants felt anxious about having children, while others were more determined to overcome related challenges. The first group was mostly represented by males, whereas girls were more optimistic about the possibility of building a family with children, although this difference has not been confirmed by other studies on the general population.

The theme of potential infertility linked to a genetic condition is a classic critical point [37,38] which, despite ongoing medical improvement in reproductive technologies [39], remains a major cause of uncertainty.

Pregnancy may represent a crucial aspect for women affected by IMDs: it can be a complex period requiring careful and specialized management. It is essential that pregnant women with an inherited metabolic disease are carefully monitored during pregnancy, because of a higher risk of complications or the necessity of a different management of the underlying disorder, such as a change in pharmacotherapy and/or diet regimen, and an intensification in the monitoring rate [40,41,42]. In this context, a particular role is played by PKU, as suboptimal metabolic control, particularly in the first trimester, may be the cause of a plurimalformative syndrome of the foetus: the so-called Maternal PKU Syndrome [43,44]. It is advisable to plan a pregnancy in a timely manner and under the supervision of specialists. This may include counselling and meetings with the specialized team before conception, to be sure that satisfactory Phe level control is reached. The optimistic consideration of a future pregnancy from our patients, albeit with the awareness of needing adjunctive support in this period, is confirmative of the common experience of the willingness and determination of PKU women to follow a very restrictive diet to ensure a positive newborn outcome [27,28].

In the background, there is the common desire of participants for further knowledge and the awareness that many aspects of this theme deserve more information. Unsurprisingly, websites were the preferred source among the younger population. In current clinical practice, healthcare teams are increasingly faced with interpreting and responding to information patients obtain from online sources. It is essential to ensure that such information is accurate and critically evaluated with the help of experts.

This study has some limitations: the sample size is small and did not allow for distinction among different types of IMDs; however, this is expected considering that IMDs are rare diseases. Furthermore, the population was heterogeneous in terms of their background and type of treatment, which are both features that can influence partner choices and family perspectives.

## 5. Conclusions

Uncertainty surrounding relationships, potential infertility and pregnancy management was common among the participants. While uncertainty may not be easily resolved, a system that includes families and healthcare teams can aid in navigating these complex decisions, which, in some cases, may start in early adolescence. Emotional support and practical assistance, with contraceptive advice, should be offered to young people with IMDs, helping them to face the complexities of relationships and family planning.

## Figures and Tables

**Figure 1 pediatrrep-17-00056-f001:**
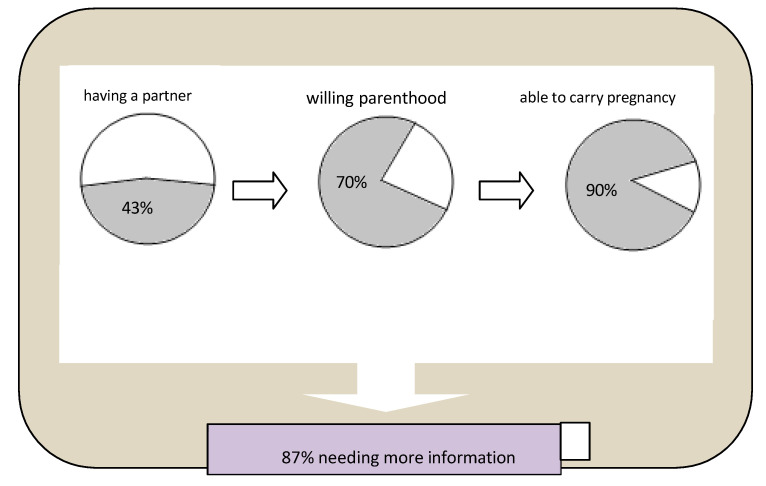
Summary representation of interviews results. **Legend:**

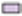

**Yes**

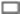

**No**.

**Table 1 pediatrrep-17-00056-t001:** Sample of interview questions according to themes.

Theme 1: Partners	Are you currently engaged/married?
	Have you ever had partners in the past?
	Are you interested in a couple’s relationship?
Theme 2: Parenthood	Have you ever thought about the possibility of becoming a parent?
	Do you think your condition could limit your ability to have children?
	Are you afraid that your child might have the same disease as you?
Theme 3: Pregnancy	Do you think you have more problems during pregnancy than your peers?
	Do you think special assistance is appropriate for your disease?
Theme 4: Information	Would you like to receive more information about fertility and parenting in the case of your disease?
	By which means would you prefer to be informed?

**Table 2 pediatrrep-17-00056-t002:** Participants’ main demographic and disease characteristics.

Age (mean ± SD)	24 ± 5.2	
Gender (F/M)	10/13	
Type of disease	PKU	13
	UCD	5
	LCFAOD	2
	GSD type 1	1
	Galactosemia	1
	MSUD	1
Treatment options	Diet	10
	Pharmacotherapy	4
	Diet + pharmacotherapy	7
	Liver transplantation	2
Genetic confirmation	22/23	

**Legend:** PKU, phenylketonuria; UCD, urea cycle disorder; LCFAOD, long-chain fatty acid oxidation disorders; GSD type 1, glycogen storage disease, type 1; MSUD, maple syrup urine disease.

## Data Availability

The data that support the findings of this study are available on request from the corresponding author (A.T.). The data are not publicly available because they contain information that could compromise the privacy of the research participants.

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
