# Peer review of "Exploring Partners, Parenting and Pregnancy Thinking in Late Adolescents and Young Adults with Inherited Metabolic Disorders"

_pediatrrep, 2025, doi:10.3390/pediatric17030056_

Round 1
Reviewer 1 Report
Comments and Suggestions for Authors
The contribution presents a cross-sectional study on thoughts about partners' choices, parenthood and pregnancy among adolescents and young adults affected by IMDs.
Surely the study, although exploratory and narrative nature, and despite the small sample size, has great relevance, considering the lack of studies on representations, thoughts, and beliefs of people affected by genetic diseases, especially young people.
However, there are some aspects that make the contribution fragile and need to be addressed:
- The study involved young people aged 16 to 35 years; this wide age range presents a critical issue; in fact the representations one’s future an be very different between adolescents and youngs. In this sense, more detailed data should be reported to compare findings between younger participants and older ones. Furthermore, gender is a very important variable in defining fears, concerns, and expectations for one's future; even here, the data, although descriptive in nature, lack any reflection on comparisons between males and females. These analyses need to be done.
- Additionally, there is no information of any kind about socio-demographic variables (e.g., work, education level, birth order, etc.), while it would have been helpful to consider them in the content analysis of the interviews.
- The introduction should be enriched with literature on representations, emotions, and needs of adolescents and young adults with genetic conditions.
- And then, what are the hypotheses of the study?
- How was the content analysis performed? With what method? Using what coding scheme?
- Clinical implications should also be explored further.
In summary, despite the interesting nature of the study, there are several methodological weaknesses that need to be resolved for the work to be accepted for publication.
Author Response
The contribution presents a cross-sectional study on thoughts about partners' choices, parenthood and pregnancy among adolescents and young adults affected by IMDs.
Surely the study, although exploratory and narrative nature, and despite the small sample size, has great relevance, considering the lack of studies on representations, thoughts, and beliefs of people affected by genetic diseases, especially young people.
R: Thank you for this comment on the usefulness of this study, we appreciate that
However, there are some aspects that make the contribution fragile and need to be addressed:
The study involved young people aged 16 to 35 years; this wide age range presents a critical issue; in fact the representations one’s future an be very different between adolescents and youngs. In this sense, more detailed data should be reported to compare findings between younger participants and older ones. Furthermore, gender is a very important variable in defining fears, concerns, and expectations for one's future; even here, the data, although descriptive in nature, lack any reflection on comparisons between males and females. These analyses need to be done.
R: Dear reviewer, we reported the age range of 16-35 as inclusion critera, however the youngest patient was aged 18, as reported, for this reason we have now changed the paper title into: “Exploring Partners, Parenting and Pregnancy Thinking of late Adolescents and Young Adults with Inherited Metabolic Disorders”
We also agree that more information on gender and age could improve interpretation of results. We have now added these information at the paragraph “Theme 1: Partners” and “Theme 4: Information”and included them in the discussion. These information were already reported in the paragraph “Theme 2: Parenthood”
Additionally, there is no information of any kind about socio-demographic variables (e.g., work, education level, birth order, etc.), while it would have been helpful to consider them in the content analysis of the interviews.
R: Thank you for pointing out the importance of these information. We had reported already this information in the Results paragraph, now information about gender have also been added.
The introduction should be enriched with literature on representations, emotions, and needs of adolescents and young adults with genetic conditions.
R: We agree with this point, and, although evidences on this topic are very scant in the literature, we have now added more studies, reporting needs, perception and choices of late adolescents and young adults affected by genetic disorders regarding partners and parenthood choices.
And then, what are the hypotheses of the study?
R: The hypothesis behind the study has now been clearly written, thank you for this input.
How was the content analysis performed? With what method? Using what coding scheme?
R: In the “ Data collection” paragraph, information about data retrieved from participants, type of coding used to anonymize patients, measures of the descriptive statistics used in the study have been added. Thank you
Clinical implications should also be explored further.
R: The main purpose of our study was not to associate the themes of partner, parenthood and pregnancy to a specific IMD, but to give a picture of such thoughts in this group of patients. The clinical implications of such considerations would require a larger number of patients to allow analysis by subgroups. This assessment, which is very interesting indeed, will be developed in a later study.
In summary, despite the interesting nature of the study, there are several methodological weaknesses that need to be resolved for the work to be accepted for publication.
R: We hope that the improvements introduced thanks to the reviewers comments and questions have significantly improved the manuscript. We also provided a deep revision of English language.
Reviewer 2 Report
Comments and Suggestions for Authors
In the paper: Abstract: Introduction The psychosocial impact of living with an Inherited Metabolic Disorder (IMB) is becoming increasingly relevant and can have a significant impact on planning the future, conditioning the reproductive decisions making made during adolescence 14 and young adulthood.
This intro sentence is disjointed maybe:
The psychosocial impact of living with an Inherited Metabolic Disorder (IMD) is becoming increasingly relevant and can have a significant impact on planning the future for young individuals. Adolescences, 14 and younger, may find themselves thinking about the reproductive impact of having children of their own.
The rest is ok.
The body of the paper:
Why did you list sentences?
Inherited metabolic disorders (IMDs) are a group of genetic conditions due to defects in specific enzymes or proteins.
IMDs can affect various metabolic pathways, including those responsible for brea ing down proteins, fats, and carbohydrates, as well as synthesizing essential molecules 35 such as hormones or neurotransmitters[1],
IMDs were first described by Sir Garrod in 1901 [2] and nowadays increasingly recognised due to improved diagnostic technologies [3-5] and to worldwide newborn screen-38 ing programmes [6].
Maybe write out as:
Inherited metabolic disorders (IMDs) are a group of genetic conditions due to defects in specific enzymes of proteins. They can affect various metabolic pathways, including those responsible for breaking down proteins, fats, and carbohydrates, as well as synthesizing essential molecules such as hormones or neruotransmitters (1). They were first described by Sir Garrod in 1901 (2) and nowadays are increasingly recotgnized due to improved diagnostic technologies (3-5) and to worldwide newborn screening programmes (6).
The current literature offers significant evidence on improved treatment options for a large number of IMDs [7,8], of which the majority were considered untreatable until a few decades ago. However, the increased survival and the improved quality of life for these patients, have raised the attention on new challenges in their management [9-11].
Final sentence of this part:
The purpose of this study was to explore thoughts about partners, parenthood and pregnancy, as well as expanding the need and preferred way of providing information among adolescents and young patients affected by different forms of IMDs.
Then to this: Material and Methods
Inclusion criteria were: males and females followed-up for IMD, between the ages of 16–35 years. Low cognitive level, mild forms of the diseases, not requiring any treatment, irregular follow-up, diagnosis not confirmed by genetic analysis represented exclusion criteria. Participants were recruited by an invitation during the regular outpatient visit. After their acceptance, an informed consent has been obtained and, a telephone interview was performed.
Maybe:
Inclusion criteria were: males and females followed up for IMDs between the ages of 16-35 years. Exclusion criteria were: low cognitive level, mild forms of the diseases that did not require any treatment, irregular follow-up, diagnosis not confirmed by genetic analysis. Participants were recruited by an invitation during the regular outpatient visit. After their acceptance, an informed consent was obtained and a telephone interview was performed.
Question: Had you thought of an exclusion property being the ability to conceive as it related to their specific IMD?
Under data collection it was stated that the survey included 22 questions that were divided into 4 thematic areas aimed at gathering information----you listed the themes in a table
You have a wide age range of 16-35—I would imagine those at least over 22 might have already been faced with this issue? I would also think they might have already experienced these decisions? Was there a difference in answers from 16-22 and 22-35?
Did you ask if they had already experienced these life events?
Results
32 were contacted and 5 did not reply and one was not interested. So did that leave 26 people to survey? Because you said 23 were eligible. Later in this section, line 114, you stated 26 people??
Could you clarify your data here?? Because in the table you list gender as F/M = 10/13 and that is 23.
Your graphical representation is nice, but I would like to see numbers under each representation to help with clarity or for added emphasis.
The discussion is pertinent and I agree this is an important topic because I’m not too sure many physicians ever think to discuss this with their patients. In fact, I would add an recommendation that the possibility of reproduction should begin rather early for children because there are children younger that 12 having sex. I’ve taken care of them.
Otherwise, except for one or two questions and my changing some language in areas, I think this is appropriate and significant to publish.
Comments on the Quality of English Language
In the paper: Abstract: Introduction The psychosocial impact of living with an Inherited Metabolic Disorder (IMB) is becoming increasingly relevant and can have a significant impact on planning the future, conditioning the reproductive decisions making made during adolescence 14 and young adulthood.
This intro sentence is disjointed maybe:
The psychosocial impact of living with an Inherited Metabolic Disorder (IMD) is becoming increasingly relevant and can have a significant impact on planning the future for young individuals. Adolescences, 14 and younger, may find themselves thinking about the reproductive impact of having children of their own.
The rest is ok.
The body of the paper:
Why did you list sentences?
Inherited metabolic disorders (IMDs) are a group of genetic conditions due to defects in specific enzymes or proteins.
IMDs can affect various metabolic pathways, including those responsible for brea ing down proteins, fats, and carbohydrates, as well as synthesizing essential molecules 35 such as hormones or neurotransmitters[1],
IMDs were first described by Sir Garrod in 1901 [2] and nowadays increasingly recognised due to improved diagnostic technologies [3-5] and to worldwide newborn screen-38 ing programmes [6].
Maybe write out as:
Inherited metabolic disorders (IMDs) are a group of genetic conditions due to defects in specific enzymes of proteins. They can affect various metabolic pathways, including those responsible for breaking down proteins, fats, and carbohydrates, as well as synthesizing essential molecules such as hormones or neruotransmitters (1). They were first described by Sir Garrod in 1901 (2) and nowadays are increasingly recotgnized due to improved diagnostic technologies (3-5) and to worldwide newborn screening programmes (6).
The current literature offers significant evidence on improved treatment options for a large number of IMDs [7,8], of which the majority were considered untreatable until a few decades ago. However, the increased survival and the improved quality of life for these patients, have raised the attention on new challenges in their management [9-11].
Final sentence of this part:
The purpose of this study was to explore thoughts about partners, parenthood and pregnancy, as well as expanding the need and preferred way of providing information among adolescents and young patients affected by different forms of IMDs.
Then to this: Material and Methods
Inclusion criteria were: males and females followed-up for IMD, between the ages of 16–35 years. Low cognitive level, mild forms of the diseases, not requiring any treatment, irregular follow-up, diagnosis not confirmed by genetic analysis represented exclusion criteria. Participants were recruited by an invitation during the regular outpatient visit. After their acceptance, an informed consent has been obtained and, a telephone interview was performed.
Maybe:
Inclusion criteria were: males and females followed up for IMDs between the ages of 16-35 years. Exclusion criteria were: low cognitive level, mild forms of the diseases that did not require any treatment, irregular follow-up, diagnosis not confirmed by genetic analysis. Participants were recruited by an invitation during the regular outpatient visit. After their acceptance, an informed consent was obtained and a telephone interview was performed.
Question: Had you thought of an exclusion property being the ability to conceive as it related to their specific IMD?
Under data collection it was stated that the survey included 22 questions that were divided into 4 thematic areas aimed at gathering information----you listed the themes in a table
You have a wide age range of 16-35—I would imagine those at least over 22 might have already been faced with this issue? I would also think they might have already experienced these decisions? Was there a difference in answers from 16-22 and 22-35?
Did you ask if they had already experienced these life events?
Results
32 were contacted and 5 did not reply and one was not interested. So did that leave 26 people to survey? Because you said 23 were eligible. Later in this section, line 114, you stated 26 people??
Could you clarify your data here?? Because in the table you list gender as F/M = 10/13 and that is 23.
Your graphical representation is nice, but I would like to see numbers under each representation to help with clarity or for added emphasis.
The discussion is pertinent and I agree this is an important topic because I’m not too sure many physicians ever think to discuss this with their patients. In fact, I would add an recommendation that the possibility of reproduction should begin rather early for children because there are children younger that 12 having sex. I’ve taken care of them.
Otherwise, except for one or two questions and my changing some language in areas, I think this is appropriate and significant to publish.
Author Response
In the paper: Abstract: Introduction The psychosocial impact of living with an Inherited Metabolic Disorder (IMB) is becoming increasingly relevant and can have a significant impact on planning the future, conditioning the reproductive decisions making made during adolescence 14 and young adulthood.
This intro sentence is disjointed maybe:
The psychosocial impact of living with an Inherited Metabolic Disorder (IMD) is becoming increasingly relevant and can have a significant impact on planning the future for young individuals. Adolescences, 14 and younger, may find themselves thinking about the reproductive impact of having children of their own.
R: Text modification made, thank you for the suggestion
The body of the paper:
Why did you list sentences?
Inherited metabolic disorders (IMDs) are a group of genetic conditions due to defects in specific enzymes or proteins.
IMDs can affect various metabolic pathways, including those responsible for brea ing down proteins, fats, and carbohydrates, as well as synthesizing essential molecules 35 such as hormones or neurotransmitters[1],
IMDs were first described by Sir Garrod in 1901 [2] and nowadays increasingly recognised due to improved diagnostic technologies [3-5] and to worldwide newborn screen-38 ing programmes [6].
Maybe write out as:
Inherited metabolic disorders (IMDs) are a group of genetic conditions due to defects in specific enzymes of proteins. They can affect various metabolic pathways, including those responsible for breaking down proteins, fats, and carbohydrates, as well as synthesizing essential molecules such as hormones or neruotransmitters (1). They were first described by Sir Garrod in 1901 (2) and nowadays are increasingly recotgnized due to improved diagnostic technologies (3-5) and to worldwide newborn screening programmes (6).
The current literature offers significant evidence on improved treatment options for a large number of IMDs [7,8], of which the majority were considered untreatable until a few decades ago. However, the increased survival and the improved quality of life for these patients, have raised the attention on new challenges in their management [9-11].
Final sentence of this part:
The purpose of this study was to explore thoughts about partners, parenthood and pregnancy, as well as expanding the need and preferred way of providing information among adolescents and young patients affected by different forms of IMDs.
R: Text modified, thank you
Then to this: Material and Methods
Inclusion criteria were: males and females followed-up for IMD, between the ages of 16–35 years. Low cognitive level, mild forms of the diseases, not requiring any treatment, irregular follow-up, diagnosis not confirmed by genetic analysis represented exclusion criteria. Participants were recruited by an invitation during the regular outpatient visit. After their acceptance, an informed consent has been obtained and, a telephone interview was performed.
Maybe:
Inclusion criteria were: males and females followed up for IMDs between the ages of 16-35 years. Exclusion criteria were: low cognitive level, mild forms of the diseases that did not require any treatment, irregular follow-up, diagnosis not confirmed by genetic analysis. Participants were recruited by an invitation during the regular outpatient visit. After their acceptance, an informed consent was obtained and a telephone interview was performed.
R: Text modified accordingly, thank you for the suggestion
Had you thought of an exclusion property being the ability to conceive as it related to their specific IMD?
R: This is a very interesting point, however, the metabolic disorders included in the study do not involve fertility issues, except for Classical Galactosemia, which can cause ovarian insufficiency. This is not the case for the interviewed patient, as she has always had good metabolic control and normal evaluations in terms of hormonal function.
Under data collection it was stated that the survey included 22 questions that were divided into 4 thematic areas aimed at gathering information----you listed the themes in a table
You have a wide age range of 16-35—I would imagine those at least over 22 might have already been faced with this issue? I would also think they might have already experienced these decisions? Was there a difference in answers from 16-22 and 22-35?
Did you ask if they had already experienced these life events?
R: Dear reviewr, the age range can be a crucial point in this context, however, our youngest patient was aged 18 years and not 16 (minimum age for inclusion). For this reason, wee have modified the title into “Exploring Partners, Parenting and Pregnancy Thinking of Late Adolescents and Young Adults with Inherited Metabolic Disorders. In this context, those who had not a partner at the time of the survey, had a mean age of 23,4 years only slight lower than those who declared a partnership whose mean age was 25,3 years (data now included in the paper ). Furthermore, mean age of those who declared to be not interested in becoming parent in the near future was 24,4 ±4 years, indicating that oldest age was not an index of propensity toward partnership and family.
Results
32 were contacted and 5 did not reply and one was not interested. So did that leave 26 people to survey? Because you said 23 were eligible. Later in this section, line 114, you stated 26 people??
Could you clarify your data here?? Because in the table you list gender as F/M = 10/13 and that is 23.
R: We thank you for highlighting this point, there was a mistake in line 169 and we confirm that sample was made by 23 patients. We apologize for that.
Your graphical representation is nice, but I would like to see numbers under each representation to help with clarity or for added emphasis.
R: Dear reviewer, we have now added percentages to the graph (see Figure 1).
The discussion is pertinent and I agree this is an important topic because I’m not too sure many physicians ever think to discuss this with their patients. In fact, I would add an recommendation that the possibility of reproduction should begin rather early for children because there are children younger than 12 having sex. I’ve taken care of them.
Otherwise, except for one or two questions and my changing some language in areas, I think this is appropriate and significant to publish.
R: Thank you for your helpful suggestions and comments, we have introduced into the discussion the importance of addressing the topic of sexuality from the early adolescence (lines 279-281).